# Improving Gamma Ray Shielding Behaviors of Polypropylene Using PbO Nanoparticles: An Experimental Study

**DOI:** 10.3390/ma15113908

**Published:** 2022-05-31

**Authors:** Ahmed M. El-Khatib, Thanaa I. Shalaby, Ali Antar, Mohamed Elsafi

**Affiliations:** 1Physics Department, Faculty of Science, Alexandria University, Alexandria 21511, Egypt; elkhatib60@yahoo.com; 2Department of Medical Biophysics, Medical Research Institute, Alexandria University, Alexandria 21561, Egypt; th_shalaby@yahoo.com (T.I.S.); newoten84@yahoo.com (A.A.)

**Keywords:** polypropylene, PbO nanoparticles, SEM, mechanical, radiation shielding

## Abstract

Recently, polymers have entered into many medical and industrial applications. This work aimed to intensively study polypropylene samples (PP) embedded with micro and nanoparticles of PbO for their application in radiation shielding. Samples were prepared by adding 10%, 30%, and 50% by weight of PbO microparticles (mPbO) and adding 10% and 50% PbO nanoparticles (nPbO), in addition to the control sample (pure polypropylene). The morphology of the prepared samples was tested; on the other hand, the shielding efficiency of gamma rays was tested for different sources with different energies. The experimental linear attenuation coefficient (LAC) was determined using a NaI scintillation detector, the experimental results were compared with NIST-XCOM results, and a good agreement was noticed. The LAC was 0.8005 cm^−1^ for PP-10%nPbO and 0.6283 cm^−1^ for PP-10%mPbO while was 5.8793 cm^−1^ for PP-50%nPbO and 3.9268 cm^−1^ for PP-50%mPbO at 0.060 MeV. The LAC values have been converted to some specific values, such as half value layer (HVL), mean free path (MFP), tenth value layer (TVL), and radiation protection efficiency (RPE) which are useful for discussing the shielding capabilities for gamma-rays. The results of shielding parameters reveal that the PP embedded with nPbO gives better attenuation than its counterpart pp embedded with mPbO at all studied energies.

## 1. Introduction

Radiation is present in a variety of forms that we encounter every day. One may be exposed to the natural radiation background, including terrestrial radiation (0.21 mSv) and cosmic radiation (0.33 mSv) [1,2,3]. Individuals that work with applications of nuclear technology—such as nuclear power plants, various radiology departments, and oncology centers—may be exposed to some additional radiation doses to such artificial radiation sources beyond those which occur naturally [4,5]. Furthermore, there is a wide usage of nuclear technology in space exploration, modification and identification of materials, coating, food sterilization, agriculture, industry, and nuclear power facilities. Therefore, researchers have studied and developed many new shielding materials to absorb radiation and protect from the hazard of ionizing radiation [6,7].

A recent development in nanotechnology has helped to change the medical rules used to prevent, diagnose, and treat diseases, and we are living in the era of nanomedical technology, for example, the manufacture of certain types of clothes that contain nanoparticles such as lead and bismuth, and these clothes are resistant to radiation in medical centers and industrial environments [8,9]. These nanoparticles are also added as a combination with concrete and mortar to work at a good attenuation of radiation coming from radioactive sources and radiological medical devices [10,11,12].

Using of polymer composites in attenuation of gamma rays has become an interesting field of research and it plays an important role in gamma radiation protection; in particular protection from scattering radiation from materials along the path of photon source. In order to study the behavior and the performance of polymer composites in radiation protection application, it is important to identify the total attenuation cross-section which are the basic parameters for the determination of the penetration depth of the photon in any material [13,14,15,16].

Polypropylene composites are reinforced by metal oxides such as Pbo which is the most used filler in polymeric matrix to shield gamma rays due to its high density and high atomic number compared to other metal oxides [17,18]. In this work, polymer composites of high density polypropylene were prepared via filling with power bulk lead and nano lead oxide nanoparticles with different filler weight percentages. The role of polymer was to acquire plasticity, easy formability, and to provide load-stress transfer. A study was designed to evaluate the ability of PP-PbO NPs versus PP-PbO in attenuation of gamma rays.

## 2. Materials

### 2.1. Polypropylene (PP)

Polypropylene is an economical material that offers a combination of outstanding physical, chemical, mechanical, thermal, and electrical properties not found in any other thermoplastic material. Compared with low- or high-density polyethylene, it has a lower impact strength, but superior working temperature and tensile strength. Its features are light weight, high tensile strength, impact resistance, high pressure resistance, excellent insulating properties, and non-toxicity. Its density ranges from 0.901 to 0.905 g/cm^−3^, tensile strength is 4800 psi, tensile modulus is 195,000 psi, tensile elongation at yield is about 12%, the compressive strength is 7000 psi, and the Rockwell hardness test yielded a 92 [19,20].

### 2.2. Lead Oxide (PbO)

In this work, micro- and nano-sized lead oxide particles were used as fillers. Microparticles were purchased locally from Abico Pharmaceuticals, with a purity of 99.7% and an average size of about 50 μm, while nanoparticles were purchased from Nano Tech Company, as they were chemically prepared. Transmission electron microscopy (TEM) [JEM-2100F, JEOL, Tokyo, Japan] at 200 kV as well as the XRD was performed on Wide Angle X-ray Diffraction with Small Angle Capability at Egyptian Nanotechnology Center (EGNC) as indicated in Figure 1. By examining these characteristics, it was confirmed that PbO nanoparticles range in size from 50.7 nm to 19.5 nm with an average size of 30 nm [21].

### 2.3. Polymer Mix Design

The samples in this study were prepared using a pressure-molding method for all polymer samples as shown in Table 1. First, a 0.0001 g sensitive electrostatic balance was used to weigh polypropylene and lead oxide, and then PP was put into a cylindrical mill at 165 °C (which is above the melting point of polypropylene) for 20 min at a rotational speed of 40 rpm. After the polypropylene was completely melted, the PbO powder, whether micro or nano, was added gradually with continuous rolling for 15 min to obtain a uniform distribution of the powder in PP. The whole mixed sample was placed in an iron frame with dimensions of 12.5 × 12.5 × 3 cm. Then, the samples were compressed by a hydraulic heat press at a pressure of 10 MPa and a temperature of 850 °C for 15 min, the pressure was gradually raised to 20 MPa for another 15 min. The sample is kept under pressure for 30 min to gradually cool down to a temperature of 400 °C, after which the pressure sample is taken and cut into circular discs for measurement [22]. The density measured by the law density = M/V, where M represents the mass of PP sheet and V its volume.

## 3. Methodology

### 3.1. SEM Test

SEM analysis (JSM-6010LV, JEOL) was used to monitor the distribution, size, and difference of micro and nano PbO particles in PP-PbO. Images acquired from SEM at a magnification order of 5000× at 20 kV [23]. The purpose of determining the morphology of the composite samples is to clarify the distribution of both micro and nano PbO particles inside the PP.

### 3.2. Attenuation Test

Sodium iodide (NaI) scintillation detector with efficiency 15% at 0.662 MeV and different radioactive point sources were used to test the attenuation parameters of the prepared samples [24,25]. Each prepared sample was tested for three different thicknesses, 0.5, 1.5, and 2 cm, with a fixed diameter of 8 cm. Initially, the detector was calibrated using Cs-137 and Co-60 sources, then the initial counting rate (N_0_) was determined in the absence of a sample, where the source-to-detector distance was 24 cm and then a sample was placed between the detector and the source at a distance of 4 cm from the detector as shown in Figure 2 to determine the counting rate (N).The characteristics of the radioactive sources that were measured are listed in Table 2 [26,27].

To know the shielding ability of the material, the linear attenuation coefficient (LAC) was experimentally determined from the following equation [28]:(1)$$\mathrm{LAC}=\frac{1}{t}\ln\frac{N_{0}}{N}$$

The experimental results of LAC for PP-mPbO samples were compared with the results obtained from NIST XCOM. The relative deviation between the two results is calculated by
(2)$${\mathrm{Dev}}_{1}\left(\%\right)=\frac{{\mathrm{LAC}}_{\mathrm{xcom}}-{\mathrm{LAC}}_{\exp}}{{\mathrm{LAC}}_{\exp}}\times 100$$

While the relative deviation between the results of LAC of the micro and the results nano filler is determined by
(3)$${\mathrm{Dev}}_{2}\left(\%\right)=\frac{{\mathrm{LAC}}_{\mathrm{nano}}-{\mathrm{LAC}}_{\mathrm{micro}}}{{\mathrm{LAC}}_{\mathrm{micro}}}\times 100$$

The linear attenuation coefficient (LAC) is the probability of photon interaction with concrete sample per unit path-length. The half and tenth value layers (HVL and TVL) are the material thicknesses enough to reduce the gamma ray intensity by 50% and 10% of its initial intensity, respectively, while the mean free path (MFP) is defined as the average distance between two successive collisions. These parameters are calculated by the following equation [29,30]:(4)$$\mathrm{HVL}=\frac{\mathrm{LN}\left(2\right)}{\mathrm{LAC}},\mathrm{TVL}=\frac{\mathrm{LN}\left(10\right)}{\mathrm{LAC}},\mathrm{MFP}=\frac{1}{\mathrm{LAC}}$$

The radiation protection efficiency (RPE) is an important parameter for estimating the efficacy of shielding materials [31,32].
(5)$$\mathrm{RPE},\%=\left[1-\frac{N}{N_{0}}\right]\times 100$$

## 4. Results and Discussion

### 4.1. SEM Results

The samples prepared from PP-mPbO and PP-nPbO were examined using scanning electron microscopy (SEM) to determine the shape of the samples and the particle distribution inside the polypropylene in addition to their size as shown in Figure 3. It turns out that the distribution of nanoparticles is more diffuse than fine particles, the smaller the size of the PbO particles, the greater their spread, in addition to being more homogeneous inside the polymer, which makes the percentage of voids in the material decrease, and this is one of the reasons for increasing the attenuation of radiation as shown in the next section.

### 4.2. Attenuation Results

The experimental LAC were determined using a NaI scintillation detector and different point radioactive sources for micro sized polypropylene samples with different percentages of PbO added in different proportions as tabulated in Table 3. In Table 3, the experimental results were compared with NIST-XCOM results and the deviation between them were obtained. The maximum relative deviation less than 4% for all prepared samples at all different discussed energies. It is clear from Table 3 that the LAC decreases with increasing energy in all the prepared samples. For example, for the first sample (PP), the value of LAC at the lowest energy was 0.1734 cm^−1^ while at the medium energy 0.662 MeV was 0.0784 cm^−1^ and at the highest energy studied was 0.0537 cm^−1^. However, when adding lead oxide, we found an increase in the value of the LAC at energy 0.122 MeV relative to the energy before it, and that was due to the presence of the K-edge (or called k-absorpance) of lead oxide at 0.082 MeV and it remains effective after this energy at medium energies and its effect disappear again at high energies. For example, for the sample PP-m PbO30 the linear attenuation coefficient was 0.969, 1.304, and 0.333 cm^−1^ at the energy of 0.081, 0.122, and 0.244 MeV, respectively. In Table 3, the measured density was also shown experimentally by measuring the mass on the disc volume. It was found that when the percentage of PbO increased, the density increased, as the density increased from 0.903 ± 0.007 to 1.645 ± 0.009 when the percentage of lead oxide was increased to 50%.

In Table 4, the results of the LAC were presented for two polypropylene samples, to which nanoparticles of lead oxide were added at a percentage of 10% and 50%, and the results were compared with the same percentages of the micro particles shown in Table 3. It turns out that polypropylene with PbO nanoparticles gives better attenuation than its counterpart polypropylene with PbO microparticles at all studied energies. We found that, for example, the LAC was 0.8005 cm^−1^ for PP-nPbO10 and 0.6283 cm^−1^ for PP-mPbO10 while it was 5.8793 cm^−1^ for PP-nPbO50 and 3.9268 cm^−1^ for PP-mPbO50 at 0.060 MeV. However, at high energy, the deviation become lower compared with the deviation at low energy, where at 1.408 MeV the LAC was 0.1100 cm^−1^ for PP-nPbO50 and 0.0933 cm^−1^ for PP-mPbO50.

The relative deviation between the micro and nano for PP-PbO10 and PP-PbO50 samples is shown in Figure 4, we found that the deviation increases with the increase in the percentage of PbO; furthermore, at the lowest energy, the deviation was higher and it starts decreasing gradually with the increase in energy. In the PP-PbO10, the deviation was 21.52% at the energy 0.060 MeV, while the deviation was 33.21% for the material PP-PbO50 at the same energy. As for the highest studied energy (1.408 MeV), the deviation was 10.10% for PP-PbO10, while it was 15.21% for PP-PbO50.

Figure 5 describes the results of comparing the linear attenuation coefficient for the four samples PP-mPbO10, PP-nPbO10, PP-mPbO50, and PP-nPbO50 at three energies 0.060, 0.081, and 0.122 MeV as shown in Figure 5a and at other three energies 0.224, 0.662, and 1.333 MeV in Figure 5b. Figure 5 generally gives a brief comparison of the results of Table 3 and Table 4. From the results, we found that the new polypropylene sample PP-nPbO50 has a higher attenuation at all energies shown in this figure than its counterpart PP-mPbO50, and the reason is the surface area of nanoparticles is higher than that of micro-particles, which makes it more diffuse and distributed inside the polypropylene material and thus makes the interaction of photons more probable, which increases the attenuation of the material.

LAC values have been converted to some specific values, such as half value layer (HVL), mean free path (MFP), and 10th value layer (TVL), which are useful for discussing the shielding capabilities for gamma-ray photons. Figure 6 displays the HVL, MFP, and TVL at different energies for PP-mPbO10, PP-nPbO10, PP-mPbO50, and PP-nPbO50 where in Figure 6a, the TVL were calculated three energies 0.060, 0.081, and 0.122 MeV. The results showed that the lower TVL is PP-nPbO50 for energy 0.060 MeV, then the TVL is lower at energy 0.122 MeV than at energy 0.081 MeV, due to the presence of the absorption region owned by PbO, where at 0.06 MeV the tenth value was 3.665, 2.876, 0.586, and 0.392 cm for P-mPbO10, PP-nPbO10, PP-mPbO50, and PP-nPbO50, respectively. Figure 6b shows the results of the distance of photon inside the polypropylene material without collisions (MFP) at three different energies 0.224, 0.365, and 0.662 MeV. Similarly, Figure 6c shows the results of the distance the polypropylene material needed to reduce the incoming gamma ray to half initial values (HVL) at three different energies—1.173, 1.333, and 1.408 MeV.

One of the most essential factors in this study is the radiation protection efficiency (RPE). The importance of this parameter lies in the fact that it gives an indication of the trend around the possibility of using prepared polypropylene samples in radiation protection applications. It also provides information about the optimal concentration of micro and nano PbO and the difference between them in the prepared samples, which makes these samples suitable for the field of radiation protection. For this reason, we examined the RPE of our prepared samples in Figure 7. We observed that RPE increases with our transition from PP to PP-PbO50, which indicates that incorporation of Pb in polypropylene has a positive effect on RPE values and this is normal, but the novelty in this work is the comparison between micro and nanoparticles of PbO when incorporated into polypropylene. The nano samples have RPE values that are superior to that of the micro samples except at the low studied energies (0.060, 0.081, and 0.122 MeV) and when 50% of PbO is incorporated into polypropylene, the RPE values reach almost 100% as shown in Figure 7. After that, the RPE values gradually decrease with the increase in energy for all the studied samples. For example, the sample PP-nPbO50 has values of 100.00%, 99.94%, 32.86%, 21.93%, and 19.75% at energies of 0.060, 0.122, 0.662, 1.173, and 1.408 MeV, respectively.

Finally, the current results were compared with previous work similar to this work; the LAC of the current work was compared with HDPE embedded with micro and nanoparticles CdO [33]. The comparison was made at 0.662 MeV as shown in Figure 8. The results showed that the current work showed that the use of PP with micro and nano PbO is better as a matrix than HDPE with micro and nano CdO in radiation attenuation at 0.662 MeV.

## 5. Conclusions

Polypropylene (PP) samples embedded with PbO micro and nanoparticles were extensively studied for their application in radiation shielding. The morphological test was carried out using SEM for the prepared samples, and it was found that the additions of nanoparticles improve the morphological properties and reduce the voids in the polymer compared to the microparticles. On the other hand, the protection efficiency of gamma rays was tested for different sources with different energies. The experimental LAC was determined using the NaI detector, and the experimental results were compared with those of NIST-XCOM and a good agreement was observed. The results of the shielding parameters show that pp embedded with nPbO provides better attenuation than that of pp embedded with mPbO at all studied energies. The LAC was 0.8005 cm^−1^ for PP-10%nPbO and 0.6283 cm^−1^ for PP-10%mPbO while was 5.8793 cm^−1^ for PP-50%nPbO and 3.9268 cm^−1^ for PP-50%mPbO at 0.060 MeV. From these findings, we conclude that these materials can be used in many applications, including the preservation of liquid radioactive sources in plastic materials made of this polymer. In addition, it can be used as an additional protective shield on walls, doors, and windows.

## Figures and Tables

**Figure 1 materials-15-03908-f001:**
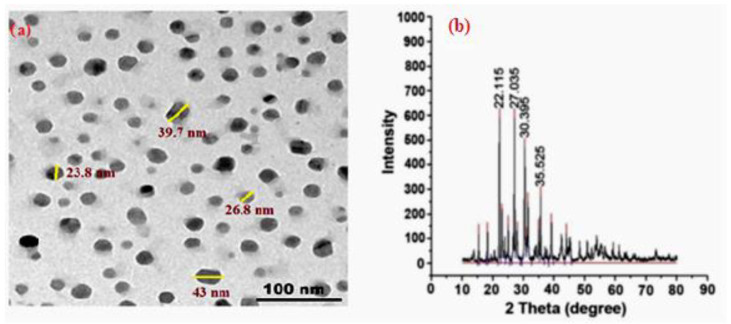
(**a**) TEM image of PbO nanoparticles, (**b**) XRD patterns of PbO nanoparticles.

**Figure 2 materials-15-03908-f002:**
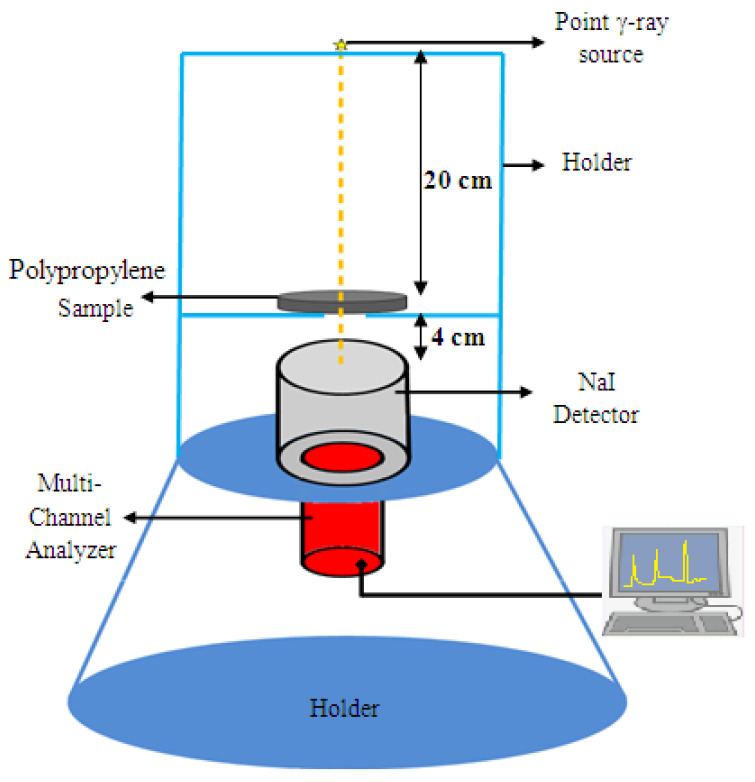
Illustration of the setup of the experimental work.

**Figure 3 materials-15-03908-f003:**
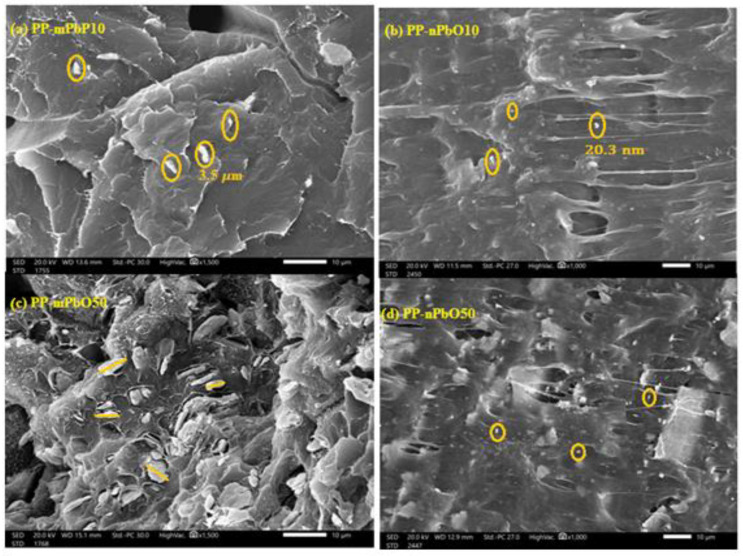
SEM images of micro and nano prepared polypropylene samples. (**a**) PP-mPbO10, (**b**) PP-nPbO10, (**c**) PP-mPbO50, and (**d**) PP-nPbO50.

**Figure 4 materials-15-03908-f004:**
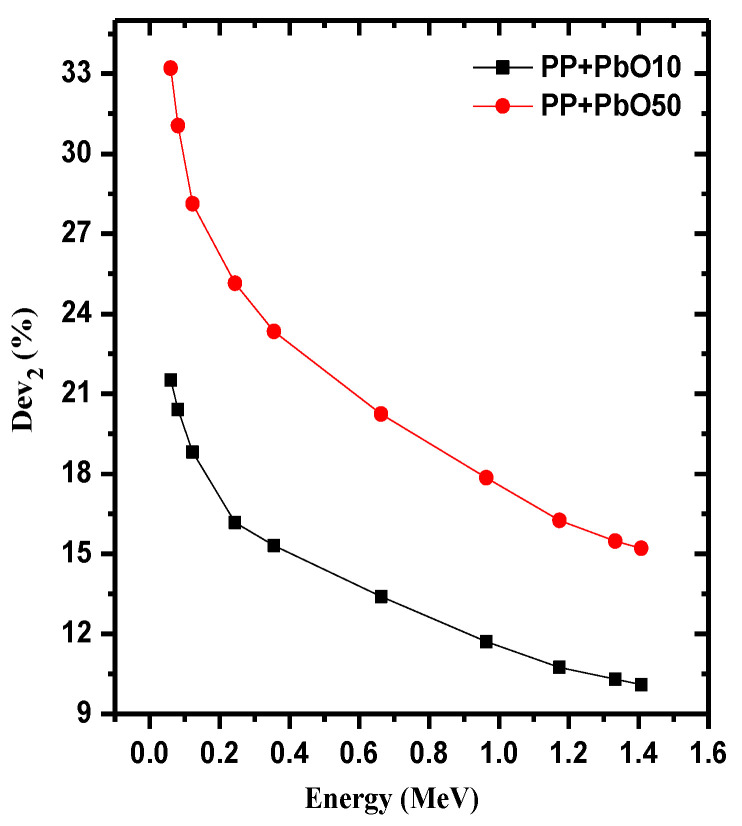
Relative deviation between the micro and nano PP samples as a function of energy.

**Figure 5 materials-15-03908-f005:**
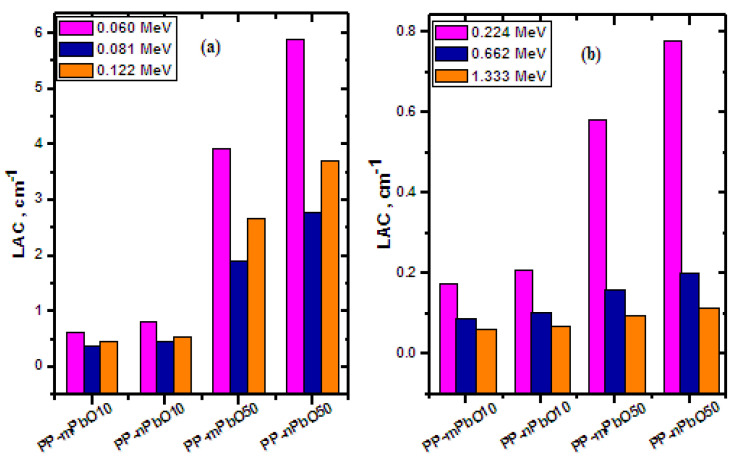
LAC of micro and nano PP samples at different energies (**a**) at 0.060, 0.081, and 0.122 MeV; and (**b**) at 0.224, 0.662, and 1.333 MeV.

**Figure 6 materials-15-03908-f006:**
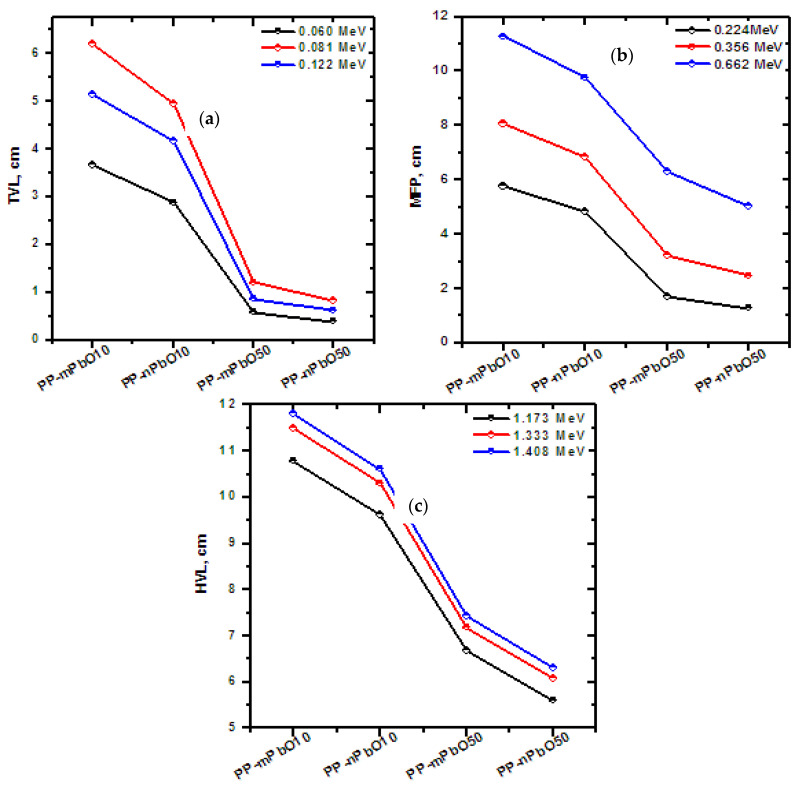
Important parameters for PP-nPbO and their corresponding materials at different energies: (**a**) TVL at 0.060, 0.081, and 0.122 MeV; (**b**) at 0.224, 0.356, and 0.662 MeV; and (**c**) 1.173, 1.333, and 1.408 MeV.

**Figure 7 materials-15-03908-f007:**
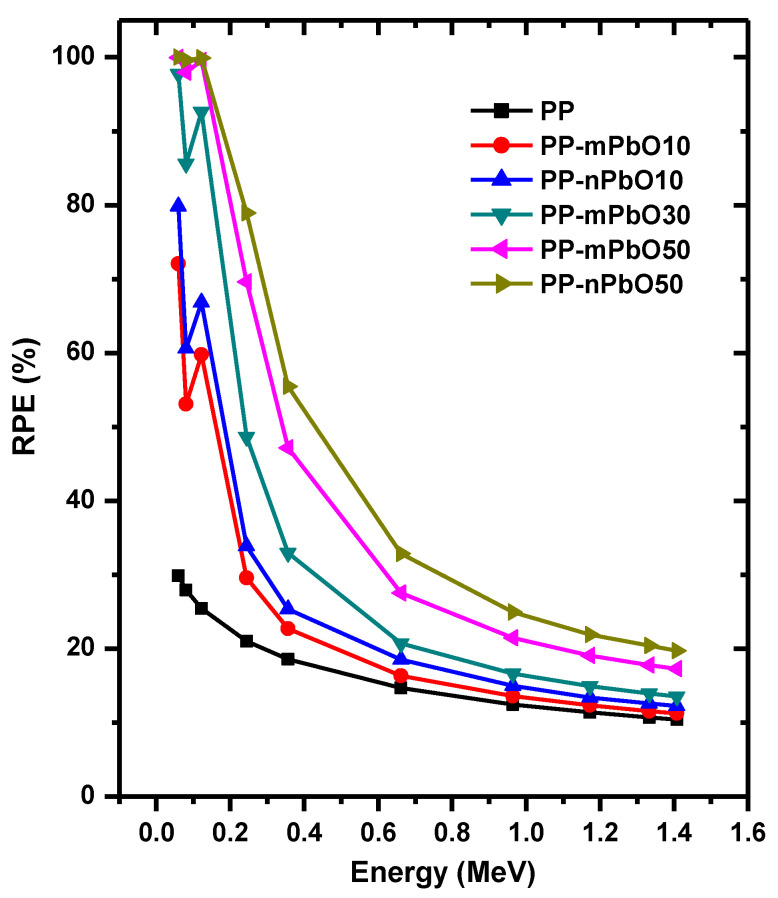
Radiation protection efficiency of all prepared samples at different energies.

**Figure 8 materials-15-03908-f008:**
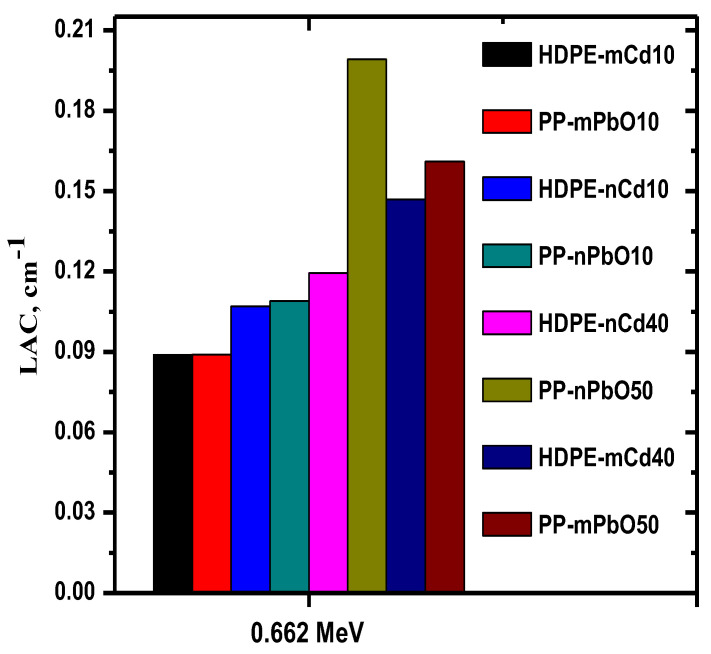
Comparison of the present LAC results with HDPE embedded with micro and nanoparticles CdO.

**Table 1 materials-15-03908-t001:** Codes, chemical compositions in weight fraction, and densities of PP-PbO composites.

Codes	Compositions (wt%)			Density (g·cm^−3^)
	PP	PbO		
		Micro	Nano	
PP	100	-		0.902 ± 0.004
PP-mPbO10	90	10		0.991 ± 0.002
PP-nPbO10	90	-	10	1.085 ± 0.009
PP-mPbO30	70	30	-	1.236 ± 0.011
PP-mPbO50	50	50	-	1.645 ± 0.007
PP-nPbO50	50	-	50	1.702 ± 0.009

**Table 2 materials-15-03908-t002:** Characteristics of the radioactive sources used in this study.

PTB Nuclide	Energy MeV	Emission Probability	Activity Bq	Uncertainty kBq
Am-241	0.060	35.9	259,000	±2.6
Cs-137	0.662	84.99	385,000	±4.0
Ba-133	0.081	32.9	275,300	±1.5
	0.356	62.05		
Co-60	1.173	99.90	212,100	±1.5
	1.333	99.982		

**Table 3 materials-15-03908-t003:** The linear attenuation coefficient of PP with PbO micro particles of different percentages and relative deviation compared with theoretical results.

Sample Code	Energy (MeV)	LAC, cm^−1^		Dev_1_ (%)	Density (g·cm^−3^)
		NIST-XCOM	Experimental		
PP	0.060	0.1775	0.1734 ± 0.0011	2.32	0.903 ± 0.007
	0.081	0.1637	0.1607 ± 0.0009	1.85	
	0.122	0.1468	0.1439 ± 0.0012	1.95	
	0.244	0.1181	0.1148 ± 0.0008	2.84	
	0.356	0.1027	0.0995 ± 0.0021	3.11	
	0.662	0.0794	0.0784 ± 0.0006	1.22	
	0.964	0.0666	0.0652 ± 0.0008	2.10	
	1.173	0.0605	0.0593 ± 0.0011	2.01	
	1.333	0.0566	0.0556 ± 0.0013	1.75	
	1.408	0.0550	0.0537 ± 0.0015	2.35	
PP-m PbO10	0.060	0.638	0.6283 ± 0.0021	1.55	0.991 ± 0.005
	0.081	0.379	0.3707 ± 0.0011	2.11	
	0.122	0.455	0.4480 ± 0.0009	1.65	
	0.244	0.176	0.1734 ± 0.0015	1.22	
	0.356	0.129	0.1237 ± 0.0019	3.85	
	0.662	0.089	0.0887 ± 0.0007	0.46	
	0.964	0.073	0.0718 ± 0.0013	1.77	
	1.173	0.066	0.0643±0.0022	2.48	
	1.333	0.062	0.0603 ± 0.0011	1.96	
	1.408	0.060	0.0587 ± 0.0018	1.77	
PP-m PbO30	0.060	1.903	1.8709 ± 0.0022	1.71	1.236 ± 0.011
	0.081	0.969	0.9486 ± 0.0012	2.11	
	0.122	1.304	1.2846 ± 0.0008	1.49	
	0.244	0.333	0.3284 ± 0.0021	1.48	
	0.356	0.200	0.1942 ± 0.0025	3.02	
	0.662	0.116	0.1149 ± 0.0005	1.10	
	0.964	0.091	0.0890 ± 0.0008	2.11	
	1.173	0.081	0.0791 ± 0.0020	2.21	
	1.333	0.075	0.0741 ± 0.0015	1.36	
	1.408	0.073	0.0719 ± 0.0025	1.38	
PP-mPbO50	0.060	4.006	3.9268 ± 0.0013	1.97	1.645 ± 0.009
	0.081	1.951	1.9115 ± 0.0032	2.02	
	0.122	2.713	2.6696 ± 0.0012	1.59	
	0.244	0.596	0.5837 ± 0.0018	2.02	
	0.356	0.319	0.3103 ± 0.0011	2.77	
	0.662	0.161	0.1589 ± 0.0020	1.35	
	0.964	0.121	0.1181 ± 0.0017	2.22	
	1.173	0.106	0.1037 ± 0.0012	2.05	
	1.333	0.098	0.0965 ± 0.0008	1.29	
	1.408	0.095	0.0933 ± 0.0016	1.57	

**Table 4 materials-15-03908-t004:** The linear attenuation coefficient of PP with 10% and 50% PbO nano particles and the relative deviation of micro samples.

Sample Code	Energy (MeV)	Experimental LAC, cm^−1^	Dev_2_ (%)	Density (g·cm^−3^)
PP-nPbO10	0.060	0.8005 ± 0.0011	21.52	1.002 ± 0.007
	0.081	0.4657 ± 0.0011	20.41	
	0.122	0.5518 ± 0.0011	18.82	
	0.244	0.2069 ± 0.0011	16.18	
	0.356	0.1461 ± 0.0011	15.31	
	0.662	0.1025 ± 0.0011	13.40	
	0.964	0.0813 ± 0.0011	11.71	
	1.173	0.0720 ± 0.0011	10.74	
	1.333	0.0673 ± 0.0011	10.30	
	1.408	0.0653 ± 0.0011	10.10	
PP-nPbO50	0.060	5.8793 ± 0.0011	33.21	1.701 ± 0.005
	0.081	2.7723 ± 0.0011	31.05	
	0.122	3.7140 ± 0.0011	28.12	
	0.244	0.7797 ± 0.0011	25.14	
	0.356	0.4047 ± 0.0011	23.33	
	0.662	0.1992 ± 0.0011	20.24	
	0.964	0.1437 ± 0.0011	17.85	
	1.173	0.1238 ± 0.0011	16.25	
	1.333	0.1142 ± 0.0011	15.48	
	1.408	0.1100 ± 0.0011	15.21	

## Data Availability

All data are available in the manuscript.

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
