# Peer review of "Improving Gamma Ray Shielding Behaviors of Polypropylene Using PbO Nanoparticles: An Experimental Study"

_materials, 2022, doi:10.3390/ma15113908_

Round 1

Reviewer 1 Report

Refer to the file as attached. 

Author Response

Thank you for your reviewing our manuscript. The attached file have the point-by-point responses to comments directed by the reviewer.

Reviewer 2 Report

This work presents an experimental study on the radiation shielding performance of PP embedded with micro and nanoparticles of PbO. After preparing several samples with various weight of particles, testing their morphology and evaluating the shielding efficiency of gamma rays, the authors conclude that PP embedded with nanoparticles PbO gives better performance. This topic is of great interest and very useful. The results and conclusion are convincing. So, I suggest the acceptance of this manuscript.

There are some minor issues:

  1. LAC in the abstract should be firstly given the full name.
  2. Why there are many sharp peaks in the XRD patterns in Fig. 1.
  3. There should be more detailed description of Captain of Fig.3.
  4. 6 is not clear and should be replotted.

Author Response

This topic is of great interest and very useful. The results and conclusion are convincing. So, I suggest the acceptance of this manuscript.

 There are some minor issues:

Reply: Thank you for the positive comments and be ensured all your suggested issues were treated point-point and modified in the manuscript.

  1. LAC in the abstract should be firstly given the full name.

Reply: Thank you for your remark. It was done.

  1. Why there are many sharp peaks in the XRD patterns in Fig. 1.

Reply: Thank you for your question. That’s normal for PbO- XRD analysis, where the sharp peak is due to the crystalline property of this powder.

  1. There should be more detailed description of Captain of Fig.3.

Reply: Thank you for your remark. It was detailed.

  1. 6 is not clear and should be replotted.

Reply: Thank you for your remark. It was clarified

Reviewer 3 Report

The article is devoted to the study of the shielding characteristics of composite materials based on polypropylene and micro- and nanoparticles of lead oxide. As research methods, X-ray diffraction and scanning electron microscopy methods were used to determine inclusions in the form of nanoparticles. Undoubtedly, the results presented by the authors are of high scientific novelty and practical significance, and are also promising for practical research. In general, the presented results of the study can be accepted for publication after the authors provide answers to all the questions raised by the reviewer during the reading of the article.

1. In the abstract, the authors need to more clearly state the purpose and relevance of this work.
2. The authors should explain what was the purpose of determining the morphology of the synthesized samples, as well as its impact on the measurement of shielding characteristics.
3. The authors should explain the choice of such different sources of gamma rays to determine the efficiency of synthesized glasses.
4. Also, the authors should explain and provide data on how evenly distributed particles of lead oxide in the polymer.
5. Shielding characteristics should be compared with the results of other works.
6. Conclusion requires significant revision.

Author Response

The article is devoted to the study of the shielding characteristics of composite materials based on polypropylene and micro- and nanoparticles of lead oxide. As research methods, X-ray diffraction and scanning electron microscopy methods were used to determine inclusions in the form of nanoparticles. Undoubtedly, the results presented by the authors are of high scientific novelty and practical significance, and are also promising for practical research. In general, the presented results of the study can be accepted for publication after the authors provide answers to all the questions raised by the reviewer during the reading of the article.

Reply: Thank you for the positive comments and be ensured all your suggested issues were treated point-point and modified in the manuscript.

  1. In the abstract, the authors need to more clearly state the purpose and relevance of this work.

Reply: Thank you for your remark. It was clarified based on your suggestion.

  1. The authors should explain what was the purpose of determining the morphology of the synthesized samples, as well as its impact on the measurement of shielding characteristics.

Reply: The purpose of determining the morphology of the composite samples is to clarify the distribution of both micro and nano PbO particles inside the PP. It turns out that the distribution of nanoparticles is more diffuse than fine particles, the smaller the size of the PbO particles, the greater their spread, in addition to being more homogeneous inside the polymer, which makes the percentage of voids in the material decrease, and this is one of the reasons for the attenuation of radiation as shown section 4.3. It was explained in the manuscript.

  1. The authors should explain the choice of such different sources of gamma rays to determine the efficiency of synthesized glasses.

Reply: Thank you for your remark. These different sources were selected to obtain a wide range of energy ranging from 0.06 to 1.401 MeV to be studied at low, medium and high energies. It was explained in the manuscript.

  1. Also, the authors should explain and provide data on how evenly distributed particles of lead oxide in the polymer.

Reply: Thank you for your remark. It was explained in the manuscript.

  1. Shielding characteristics should be compared with the results of other works.

Reply: Thank you for your remark. The comparison of our data was added in the manuscript.

  1. Conclusion requires significant revision.

Reply: Thank you for your remark. The conclusion was revised.

Round 2

Reviewer 3 Report

The authors answered all the questions posed, the article can be accepted for publication.